# PARP Inhibitors for Metastatic CRPC: More Answers than Questions, a Systematic Review and Meta-Analysis

**DOI:** 10.3390/ph18071015

**Published:** 2025-07-08

**Authors:** Ray Manneh, Javier Molina-Cerrillo, Guillermo de Velasco, Linda Ibatá, Susan Martínez, Álvaro Ruiz-Granados, Teresa Alonso-Gordoa

**Affiliations:** 1Medical Oncology Department, Sociedad de Oncología y Hematología del César, Valledupar 200001, Colombia; 2Medical Oncology Department, Hospital Universitario Ramón y Cajal, 28034 Madrid, Spain; 3Medical Oncology Department, Hospital Universitario 12 de Octubre, 28041 Madrid, Spain; 4Epidemiology Department, EpiThink Health Consulting, Bogotá 110231, Colombia

**Keywords:** metastatic castration-resistant prostate cancer, PARP inhibitors, DNA repair alterations, precision oncology

## Abstract

PARP inhibitors (PARPi), alone or in combination with androgen receptor signaling inhibitors (ARSi), have shown clinical benefit in metastatic castration-resistant prostate cancer (mCRPC), particularly in tumors with homologous recombination repair (HRR) gene alterations. Recent data from the TALAPRO-2 trial complete the current evidence on PARPi–ARSi combination strategies in this setting. **Background/Objectives**: To evaluate the efficacy and safety of PARPi-based therapies—monotherapy and combination with ARSi—in patients with mCRPC, focusing on molecular subgroups defined by DNA repair alterations. **Methods**: We conducted a systematic review and meta-analysis of phase III randomized controlled trials (RCTs) assessing PARPi as monotherapy or in combination with ARSi. Searches were performed in PubMed, EMBASE, the Cochrane Library, and oncology conference proceedings up to February 2025. Outcomes included radiographic progression-free survival (rPFS), overall survival (OS), second progression-free survival (PFS2), and grade ≥3 adverse events (AEs). Data were pooled using a random-effects model, with subgroup analyses by DNA repair status. **Results:** Five RCTs (*n* = 2921) were I confirmincluded: three on combination therapy (*n* = 2271) and two on monotherapy (*n* = 650). Combination therapy improved rPFS in the ITT (HR = 0.64; 95% CI: 0.56–0.74), HRRm (HR = 0.55; 95% CI: 0.44–0.68), and BRCAm (HR = 0.33; 95% CI: 0.18–0.58) subgroups. OS was also improved in the ITT (HR = 0.80; 95% CI: 0.70–0.92), HRRm (HR = 0.68; 95% CI: 0.55–0.83), and BRCAm (HR = 0.54; 95% CI: 0.34–0.85) groups. No benefit was observed in non-HRRm patients. PFS2 favored combination therapy (HR = 0.77; 95% CI: 0.64–0.91). Grade ≥3 AEs were more frequent (RR = 1.44; 95% CI: 1.20–1.73). Monotherapy improved rPFS in ITT (HR = 0.46; 95% CI: 0.20–0.81) and BRCAm (HR = 0.33; 95% CI: 0.15–0.75); OS benefit was seen only in BRCAm (HR = 0.73; 95% CI: 0.57–0.95). **Conclusions**: PARPi therapies improve outcomes mainly in HRR- and BRCA-mutated mCRPC. Molecular selection is key to optimizing benefit and minimizing toxicity. Further research on the activity of PARPi combinations in non-HRR mutated mCRPC is needed to better understand the underlying mechanisms of efficacy.

## 1. Introduction

Prostate cancer (PC) is the second most prevalent cancer worldwide, with an incidence of 30.7 per 100,000 men in 2020, accounting for over 1.4 million new cases annually [1]. The proportion of patients diagnosed at an advanced stage has increased in recent years and is currently estimated at 15% to 25%, with a five-year survival rate of just 29.8% [2,3,4]. Despite therapeutic advances in the metastatic castration-sensitive prostate cancer (mCSPC) setting, most patients eventually progress to metastatic castration-resistant prostate cancer (mCRPC), which is associated with poor outcomes. Available systemic treatments include androgen receptor signaling inhibitors (ARSI), radium-223, lutetium-177 PSMA, cabazitaxel, and docetaxel. However, the optimal sequencing of these agents remains uncertain.

Alterations in the DNA damage response (DDR) pathway—present in approximately 30% of mCRPC cases—render tumor cells more dependent on alternative DNA repair mechanisms and are associated with disease progression and poor prognosis [5,6,7]. These findings have led to the development of poly (ADP-ribose) polymerase (PARP) inhibitors (PARPi) as a promising therapeutic strategy. PARPis inhibit the PARP enzyme, a key component of DNA repair, and they are particularly effective in tumors with homologous recombination repair (HRR) deficiencies, including BRCA1/2 mutations [8].

Clinical trials have demonstrated the efficacy of PARPis in various settings [8]. The phase II TOPARP-A trial first established the antitumor activity of olaparib in mCRPC patients with DDR mutations, with subsequent validation in phase III trials, such as PROfound and TRITON3, which confirmed the benefit of olaparib and rucaparib in previously treated patients with DDR alterations [9,10,11,12].

More recently, combination strategies—particularly PARPi plus ARSI—have shown promise in first-line mCRPC treatment. Phase III trials, such as PROpel, TALAPRO-2, and MAGNITUDE, have explored this approach, aiming to extend clinical benefit beyond biomarker-selected populations [10,13,14,15]. These studies raise important questions regarding the optimal use of PARPis. Should treatment be limited to patients with DDR mutations, especially BRCA alterations, or could broader populations benefit? With the recent publication of final overall survival data from TALAPRO-2 [16,17], the current evidence base allows for a comprehensive reassessment of the efficacy of PARPi, both as monotherapy and in combination with ARSI, across molecular subgroups in mCRPC.

We conducted a meta-analysis of phase III randomized controlled trials evaluating the efficacy of PARPi therapies—used either as monotherapy or in combination with ARSI—in patients with mCRPC. This study aims to provide clinically relevant insights to guide therapeutic decision making.

Specifically, we sought to address the following research questions:-What are the efficacy outcomes of PARPi-based therapies—monotherapy or in combination with ARSI—across distinct molecular subgroups in mCRPC?-What are the safety outcomes of PARPi therapies in monotherapy and combination strategies?

## 2. Methods

This systematic review followed the methodology outlined in the Cochrane Handbook for Systematic Reviews of Interventions for evidence collection and analysis [18]. The study findings are reported in accordance with the Preferred Reporting Items for Systematic Reviews and Meta-Analyses (PRISMA) Statement guidelines [19]. This review was retrospectively registered in the Open Science Framework (OSF), where a summary of the protocol is available at https://doi.org/10.17605/OSF.IO/HWE9S. We extensively searched PubMed, EMBASE, and the Cochrane Library databases (the last search was updated in February 2025). Furthermore, we examined additional sources, including http://clinicaltrials.gov, abstracts, and presentations from the European Society of Medical Oncology (ESMO) and the American Society of Clinical Oncology (ASCO) conferences. The search strategy was based on the terms (‘prostate cancer’) AND (‘nicotinamide adenine dinucleotide adenosine diphosphate ribosyl-transferase inhibitor’ OR ‘parp inhibitor’) (Appendix A).

We included only phase III randomized controlled trials (RCTs) involving patients with metastatic castration-resistant prostate cancer (mCRPC) treated with PARP inhibitors as monotherapy or in combination with ARSI. The search was restricted to studies published in English or Spanish, with no limitations on publication date or status. When multiple reports of the same trial were available, only the most recent publication was considered.

The decision to exclude phase II studies was based on the need to synthesize high-level evidence from trials with robust design and adequate statistical power. While early-phase trials provided essential proof-of-concept data, this meta-analysis was designed to assess comparative efficacy and safety using phase III data exclusively to better inform clinical decision making.

The outcomes of interest were overall survival (OS), radiographic progression-free survival (rPFS), second progression-free survival (PFS2), and the incidence of grade ≥3 adverse events (AEs) according to the Common Terminology Criteria for Adverse Events (CTCAE) version 4. For the analysis of rPFS, data assessed through blinded independent central review were preferred to minimize bias.

Two researchers (R.M.K. and T.A.G.) independently examined the publications and collected the data. Discrepancies were resolved through consensus. All citations retrieved from the searches were stored in a reference database. The collected data included the author, year, population, treatment, sample size, and summarized estimates of relevant outcomes. The methodological quality of the qualifying trials was evaluated using Cochrane’s Risk of Bias (RoB) tool, categorized into three levels: high bias, low bias, and unclear [20]. The GRADE methodology was employed to assess the quality of evidence, categorizing it as high, moderate, low, or very low. This classification was determined based on bias risk, directness, precision, and consistency in treatment effects [21].

We employed a direct frequentist meta-analysis using a random-effects model, specifically, the DerSimonian and Laird method [22,23]. This approach was chosen over a fixed-effects model to account for potential clinical and methodological heterogeneity across studies, particularly given the variability in patient populations, treatment regimens, and follow-up durations. While Bayesian models offer advantages in incorporating prior information, the available data and scope of our analysis favored the more widely adopted frequentist approach for transparency and comparability. Before data pooling, the authors determined the similarity of patients and treatment characteristics, follow-up duration, and outcome definitions. Hazard ratios (HRs) and their corresponding 95% confidence intervals (CI) were pooled and presented through forest plots. A pooled relative risk was calculated for the incidence of grade ≥3 AEs. Statistical heterogeneity was assessed through Cochran’s Q test, with a significance level of *p* < 0.05, and quantified using *I*^2^ statistics [24]. Given the limited number of included studies, publication bias could not be formally assessed. All statistical analyses were performed using R software (version 4.4.2) within the RStudio integrated development environment.

The results are presented in subgroups according to the DNA repair alteration, including intention-to-treat (ITT) population, homologous recombination repair gene mutation (HRRm), BRCA mutation (BRCAm), and those without HRR mutations (non-HRRm).

## 3. Results

### 3.1. Trials Characteristics

A total of 784 records were identified through electronic database searches and manual screening. Based on predefined selection criteria, five phase III RCTs were included (see PRISMA flow diagram in Figure 1).

Of the studies included, three evaluated combination therapy with PARPis —PROpel [14], MAGNITUDE [13,25], and TALAPRO-2 [15,16,17]—involving 1136 patients in the experimental arms (receiving olaparib plus abiraterone acetate and prednisone [AAP], niraparib plus AAP, or talazoparib plus enzalutamide) and 1135 patients in the control arms (AAP or enzalutamide monotherapy). Two trials assessed PARPis monotherapy—PROfound [10,11] and TRITON3 [12]—with 432 patients in the intervention arms (treated with olaparib or rucaparib) and 218 patients in the control arms (receiving physician’s choice therapy). Key characteristics of the included studies are summarized in Table 1 and Table 2, and details of excluded studies are provided in Appendix A.

### 3.2. Quality Assessment

According to the Cochrane Collaboration Risk of Bias tool [20], the combination therapy trials were judged to have a low overall risk of bias. In contrast, the monotherapy trials were rated as having an unclear risk of performance bias, primarily due to limitations in blinding of participants and personnel, as well as incomplete reporting of allocation concealment. The risk of other biases in the monotherapy trials was also considered unclear due to insufficient information regarding protocol availability and incomplete reporting of methodological aspects. A visual summary of the risk of bias assessment is provided in Figure 2 to enhance clarity. Given the limited number of included studies, publication bias could not be formally assessed.

The certainty of the evidence for combination therapy was rated as moderate to high across most evaluated subgroups. For monotherapy, the certainty was rated as moderate, primarily due to wide confidence intervals and inclusion of the possibility of no effect in some outcome estimates.

### 3.3. Combined Therapy Efficacy

Data on rPFS and OS were extracted from three RCTs: the MAGNITUDE trial [13,25], which compared niraparib plus AAP to AAP plus placebo; the PROpel trial [14], which evaluated olaparib plus AAP versus AAP plus placebo; and TALAPRO-2 [15,16,17], which assessed talazoparib plus enzalutamide against enzalutamide plus placebo.

rPFS was consistently reported across studies for molecularly defined subgroups, and PROpel [14] and TALAPRO-2 [15,16,17] reported data for the intention-to-treat (ITT) population. A significant benefit favoring combination therapy was observed in the ITT population (HR = 0.64; 95% CI: 0.56–0.74; *I*^2^ = 0.0%), the HRRm subgroup (HR = 0.55; 95% CI: 0.44–0.68; *I*^2^ = 50.5%), and the BRCAm subgroup (HR = 0.33; 95% CI: 0.18–0.58; *I*^2^ = 70.5%). In the non-HRRm population, the combination did not reach statistical significance (HR = 0.81; 95% CI: 0.64–1.03; *I*^2^ = 43.9%) (Figure 3).

OS data were reported for ITT population in PROpel [14] and TALAPRO-2 [15,16,17], while all three trials (PROpel [14], TALAPRO-2 [15,16,17], and MAGNITUDE [13,25]) provided data for the HRRm and BRCAm subgroups. A significant benefit in OS favoring combination therapy was observed in the ITT population (HR = 0.80; 95% CI: 0.70–0.92; *I*^2^ = 0.0%), the HRRm subgroup (HR = 0.68; 95% CI: 0.55–0.83; *I*^2^ = 32.4%), and the BRCAm subgroup (HR = 0.54; 95% CI: 0.34–0.85; *I*^2^ = 61.6%). In the non-HRRm population, no statistically significant benefit was observed (HR = 0.84; 95% CI: 0.70–1.02; *I*^2^ = 0.0%) (Figure 4).

Two studies reported PFS2, defined as the time between randomization and second progression or death on next-line anticancer therapy. In the ITT population, differences in PFS2 were observed favoring combination therapy (HR = 0.77; CI 95% 0.64 –0.91; *I*^2^ = 0.0%). Figure 5 shows the results.

### 3.4. Combined Therapy Safety

Table 3 shows grade ≥3 AEs reported, occurring in >2% of ITT population, except for the MAGNITUDE trial, where safety data are only reported for the HRRm population.

Grade ≥3 AEs were more frequent in the combination therapy arms. The pooled relative risk (RR) for grade ≥3 AEs was 1.44 (95% CI: 1.20–1.73; *I*^2^ = 76.7%), favoring the control group (Figure 6). Because the MAGNITUDE trial reported safety data only for the HRRm population, we performed a sensitivity analysis excluding that study. The result remained consistent, with a pooled risk ratio (RR) of 1.44 (95% CI: 1.07–1.93), confirming the robustness of the overall estimate despite the data limitation.

### 3.5. Monotherapy Efficacy

PFS and OS data were extracted from two RCTs: the PROfound trial [10,11], which evaluated olaparib monotherapy versus enzalutamide or AAP, and the TRITON3 trial [12], which assessed rucaparib monotherapy versus enzalutamide, AAP, or docetaxel. Both studies included men with mCRPC harboring BRCA1, BRCA2, or ATM gene alterations.

In the ITT population, the pooled HR for PFS showed a statistically significant benefit in favor of PARPi (HR = 0.46; 95% CI: 0.20–0.81; *I*^2^ = 87.0%). This benefit was clear for patients with BRCA1/2 alterations (HR = 0.33; 95% CI: 0.15–0.75; *I*^2^ = 90.4%). No significant differences in PFS were observed in the ATMm subgroup (HR = 0.99; 95% CI: 0.69–1.42; *I*^2^ = 0.0%) (Figure 7).

Regarding OS, the pooled HR in the ITT population did not reach statistical significance (HR = 0.82; 95% CI: 0.61–1.11; *I*^2^ = 50.6%). In the BRCA1/2-mutated subgroup, PARPi treatments showed a significant improvement in OS (HR = 0.73; 95% CI: 0.57–0.95; *I*^2^ = 0.0%). No difference in OS was found in patients with ATMm (HR = 1.08; 95% CI: 0.74–1.58; *I*^2^ = 0.0%) (Figure 8).

### 3.6. Monotherapy Safety

Table 4 summarizes the incidence of grade ≥3 AEs reported in PROfound [10,11] and TRITON3. The incidence of AEs in the control groups varied between studies, largely due to the inclusion of docetaxel as a comparator in TRITON3.

A pooled RR was calculated to assess the incidence of grade ≥3 AEs. The meta-analysis showed a higher risk of severe AEs in the PARPi treatment arms compared to controls (RR = 1.21; 95% CI: 1.01–1.44; *I*^2^ = 24.2%). The TRITON3 trial individually did not demonstrate a significant difference in AE incidence between treatment arms, which may be attributed to the higher baseline toxicity associated with docetaxel in the control group (Figure 9).

To facilitate comparison across treatment strategies and molecular subgroups, a summary table of key outcomes—including rPFS, OS, PFS2, and grade ≥3 adverse events—for both combination therapy and monotherapy is presented in Table 5. This table provides an at-a-glance overview of treatment effects and supports the interpretation of subgroup-specific findings discussed above.

## 4. Discussion

Metastatic castration-resistant prostate cancer (mCRPC) represents the final stage of prostate cancer, and most patients will ultimately succumb to their disease [26]. Despite the approval of multiple agents with distinct mechanisms of action in recent decades, improvements in overall survival (OS) have been modest [27].

Among these therapies, PARP inhibitors (PARPis) have emerged as an important option. Olaparib and rucaparib are approved for the treatment of patients with HRR gene-mutated and BRCA1/2-mutated mCRPC based on the results of the PROfound and TRITON-3 trials [28,29]. Although trials evaluating the combination of PARPis with ARSI included unselected populations, FDA approvals have been restricted to molecularly selected patients. In 2023, the FDA approved the combination of olaparib plus abiraterone acetate and prednisone (AAP) and talazoparib plus enzalutamide for BRCA-mutated and HRR gene-mutated mCRPC, respectively [30,31]. More recently, niraparib with AAP was approved in BRCA-mutated mCRPC [32].

This systematic review and meta-analysis aimed to evaluate the efficacy and safety of PARPis, either as monotherapy or in combination with ARSI, across subgroups included in the pivotal trials. For monotherapy, we included data from the PROfound trial (Cohort A) and the TRITON-3 trial, both enrolling patients with BRCA1/2 or ATM mutations [10,12]. Despite some design differences, our results consistently demonstrated improved progression-free survival (PFS) in patients with BRCA mutations treated with PARPis, independent of prior taxane use. Conversely, no clear benefit was observed in the ATM-mutated subgroup. A significant OS benefit was also found in BRCAm patients, even with high crossover rates in TRITON-3. Grade ≥3 adverse events (AEs), particularly myelosuppression, were more frequent with PARPis compared to AAP or enzalutamide but comparable to docetaxel. These findings underscore the greater efficacy of PARPis over ARSI or docetaxel in BRCAm patients and highlight the need for alternative strategies in ATM-altered mCRPC.

For combination therapy, we analyzed data from PROpel, MAGNITUDE, and TALAPRO-2 [13,14,15]. These trials varied in prior treatment allowances, molecular stratification, and comparator arms. Notably, only MAGNITUDE restricted enrollment based on HRR mutation status. Our pooled analysis showed a significant radiographic PFS (rPFS) benefit with PARPi combinations irrespective of HRR status. However, heterogeneity was high (*I*^2^ = 70.5%), and the magnitude of benefit was substantially lower in non-BRCAm subgroups. In fact, no statistically significant benefit was observed in non-HRRm patients (HR = 0.81 for rPFS; HR = 0.84 for OS), suggesting that the benefit may be largely driven by the BRCAm subgroup. This lack of efficacy may be explained by the absence of homologous recombination deficiency in non-HRRm tumors, which limits the synthetic lethality mechanism targeted by PARP inhibition. Additionally, the non-HRRm population is molecularly heterogeneous and may include tumors with intact DNA repair capacity. However, some non-HRRm patients might still benefit from PARPi-based combinations due to alternative biological features. Preliminary translational data suggest that molecular phenotypes associated with replication stress, high androgen receptor activity, or chromatin remodeling defects could modulate sensitivity to PARPis, even in the absence of canonical HRR mutations [33]. Further prospective research is needed to better characterize these subgroups and identify novel predictive biomarkers beyond HRR status.

In terms of OS, a benefit was observed in the intention-to-treat (ITT) population from PROpel and TALAPRO-2, whereas MAGNITUDE did not report OS data for non-HRRm patients due to early futility. In terms of OS, a benefit was observed in the intention-to-treat (ITT) population from PROpel and TALAPRO-2, whereas MAGNITUDE did not report OS data for non-HRRm patients due to early futility [13]. Toxicity remains a concern, as combinations with PARPis led to more grade ≥3 AEs than ARSI alone, particularly hematologic events, highlighting the need to balance efficacy with tolerability. Notably, the MAGNITUDE trial reported safety outcomes only for the HRRm population, raising concerns about potential bias in the pooled estimate. To address this, we conducted a sensitivity analysis excluding the MAGNITUDE trial, which yielded a similar pooled relative risk (RR = 1.44; 95% CI: 1.07–1.93), confirming the robustness of our findings and reinforcing the conclusion that the combination therapy is associated with increased toxicity compared to ARSI alone.

Heterogeneity in efficacy outcomes was notably high across analyses and likely stems from several clinical and methodological differences among trials. These include variations in trial design (e.g., double-blind vs. open-label), patient selection criteria (e.g., inclusion of de novo vs. pretreated mCRPC), prior treatments allowed (e.g., use of AAP or chemotherapy in the mHSPC setting), and the definition and timing of endpoints. For example, TALAPRO-2 and MAGNITUDE permitted prior use of AAP or docetaxel in mHSPC, while PROpel did not, potentially influencing response and resistance patterns [13,14,15]. Differences in HRR gene panel composition, sequencing platforms, and central vs. local mutation testing also introduced heterogeneity in the molecular characterization of patients. Additionally, the proportion of patients harboring BRCA versus non-BRCA alterations varied across trials. These factors may explain the variability in treatment effects and contribute to the observed statistical heterogeneity. Although a formal sensitivity analysis was not feasible due to the limited number of available studies, we addressed these issues by presenting and interpreting results separately for each predefined molecular subgroup, which facilitates a more accurate understanding of treatment effects.

A key consideration is the generalizability of these results. Clinical trial populations are typically younger, have fewer comorbidities, and are subject to stricter eligibility criteria than real-world patients. This discrepancy may lead to overestimation of treatment efficacy and underestimation of toxicity in broader clinical practice. Access to molecular testing, particularly next-generation sequencing (NGS), remains limited in many settings due to cost, lack of reimbursement, and infrastructure challenges, which can delay or prevent appropriate patient selection for PARPi therapy. Moreover, real-world patients frequently present with comorbidities, such as cardiovascular disease or baseline cytopenias, which may exacerbate the risk of hematologic toxicity associated with PARPi. Additionally, a growing proportion of patients now receive intensified systemic therapy in the mHSPC setting, such as early use of ARSI or chemotherapy, which was underrepresented in clinical trials and may influence treatment sequencing and resistance mechanisms. These factors should be carefully considered by clinicians when extrapolating clinical trial findings to routine care, particularly in resource-constrained environments.

Compared with three recent meta-analyses of PARPi in mCRPC, our analysis provides several incremental insights. The meta-analysis by Messina et al. [34] focused exclusively on first-line PARPi + ARSI combinations (PROpel, MAGNITUDE, TALAPRO-2) and reported pooled hazard ratios of 0.62 for rPFS and 0.84 for OS in biomarker-unselected populations, suggesting clinical benefit regardless of HRR status. The study by Roberto et al. [35] evaluated PARPi both as monotherapy and in combination across first- and second-line settings, finding that rPFS improved in all subgroups, but OS’s benefit was significant only in HRR-mutated patients. It also found no added value of PARPi + ARSI in the second-line setting. Finally, the meta-analysis by Sayyid et al. [36] synthesized trial-level data and highlighted the superiority of PARPi in BRCA-mutated populations, but it did not incorporate updated survival data from TALAPRO-2 or the second interim analysis of MAGNITUDE. In contrast, our study includes these newer datasets, provides disaggregated estimates for BRCAm, HRRm, and non-HRRm subgroups, and conducts a sensitivity analysis excluding MAGNITUDE for safety outcomes. These additions contribute to a more comprehensive and contemporary understanding of PARPi-based treatment effects across molecular subtypes and treatment contexts.

Finally, our findings support current regulatory decisions limiting PARPi combination approvals to HRRm or BRCAm mCRPC patients. Given the limited benefit observed in non-HRRm subgroups, extending these therapies to biomarker-unselected populations may not be justified at this time. Continued translational research is needed to refine patient selection and optimize sequencing strategies for PARPi-based therapies.

## 5. Conclusions

PARP inhibitors, either as monotherapy or in combination, represent an essential therapeutic option in mCRPC, particularly for patients harboring pathogenic BRCA mutations. For other HRR alterations, a clinical benefit remains evident, albeit with reduced magnitude, likely reflecting the variable sensitivity across different genes involved in the DNA damage response. Additionally, some patients with non-HRR alterations may also derive benefit from PARPi-based combinations. Our findings reinforce the need for continued research to better elucidate the antitumor activity of PARP inhibitors and to explore additional therapeutic strategies beyond BRCA mutations. Importantly, our results highlight the critical role of molecular testing in guiding treatment selection. In clinical practice, comprehensive genomic profiling—preferably using next-generation sequencing (NGS) panels covering BRCA1/2 and other HRR genes—should be conducted at the time of diagnosis of metastatic disease or upon progression to castration resistance. While such testing is becoming more accessible, challenges remain in resource-limited settings, including cost, turnaround time, and infrastructure limitations, which warrant the development of scalable strategies to ensure equitable access to precision oncology.

## Figures and Tables

**Figure 1 pharmaceuticals-18-01015-f001:**
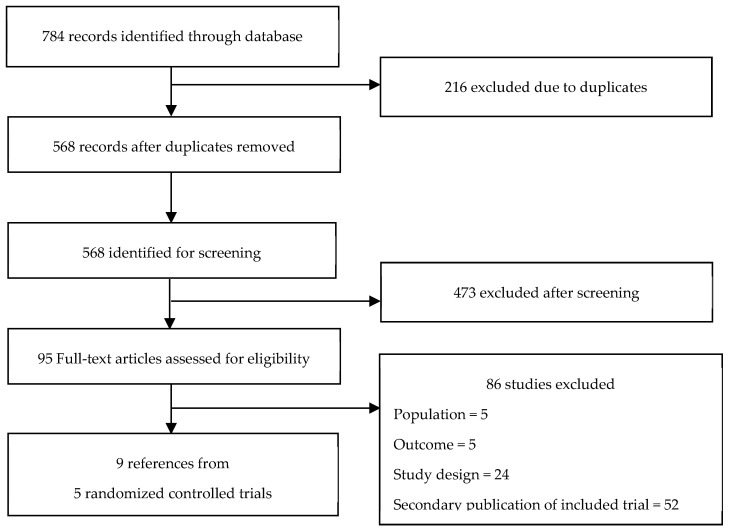
Flow diagram of the systematic review.

**Figure 2 pharmaceuticals-18-01015-f002:**
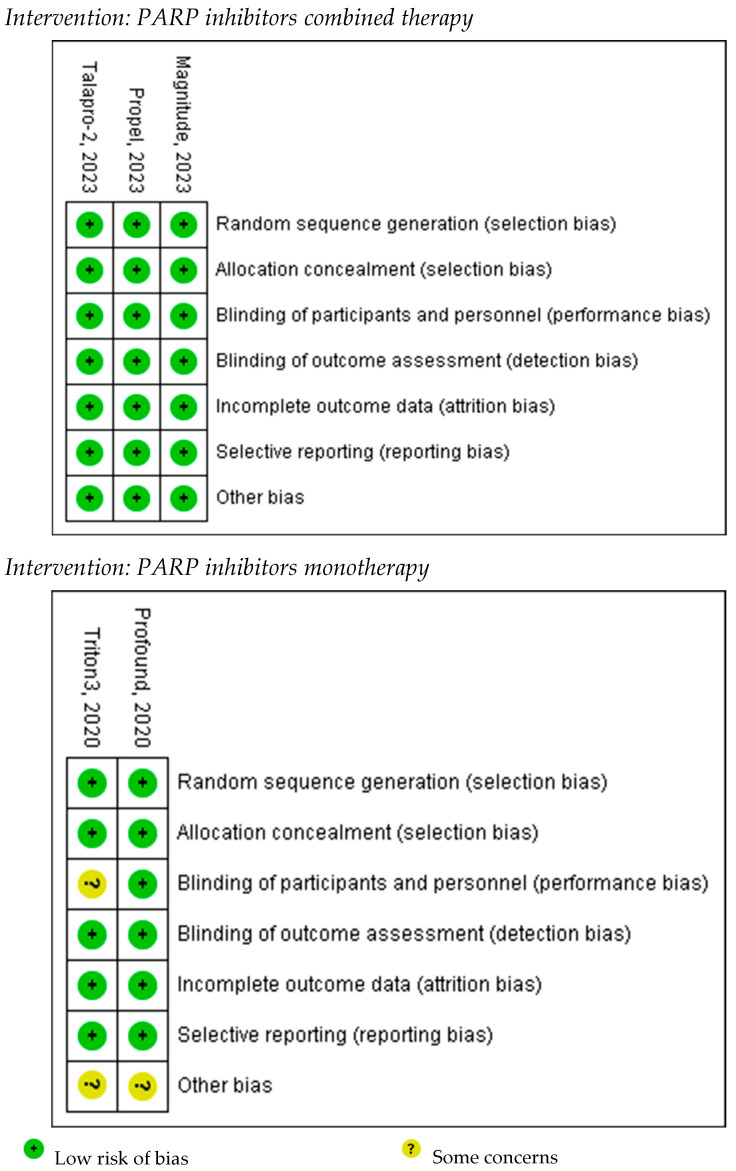
Cochrane Risk of Bias assessment summary for included RCTs.

**Figure 3 pharmaceuticals-18-01015-f003:**
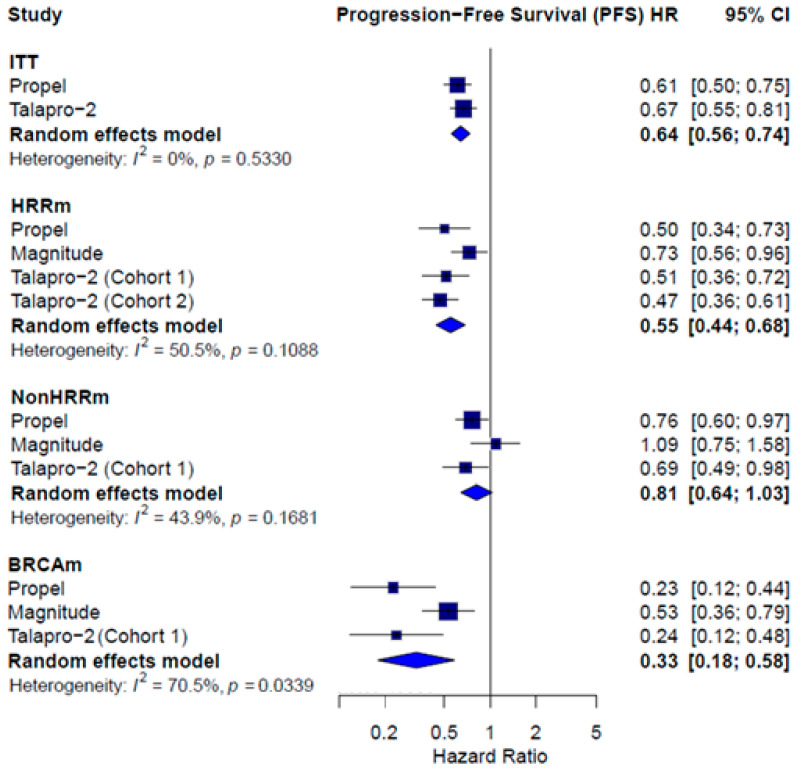
Forest plot estimating rPFS of patients treated with combined therapy with PARPi in comparison to AAP or enzalutamide plus placebo.

**Figure 4 pharmaceuticals-18-01015-f004:**
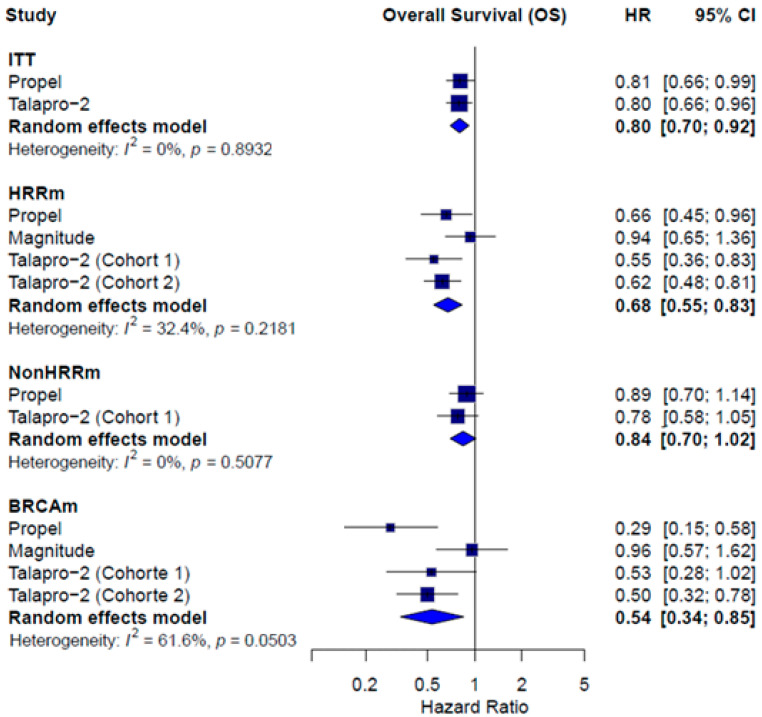
Forest plot estimating OS of patients treated with combined therapy with PARPi in comparison to AAP or enzalutamide plus placebo.

**Figure 5 pharmaceuticals-18-01015-f005:**
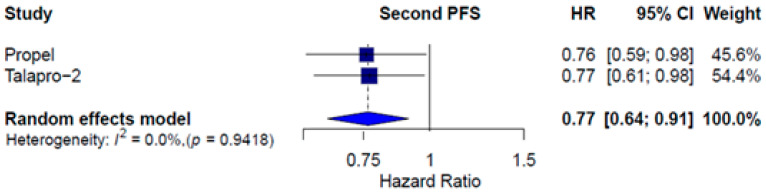
Forest plot estimating second PFS of patients treated with combined therapy with PARPi compared to AAP or enzalutamide plus placebo.

**Figure 6 pharmaceuticals-18-01015-f006:**
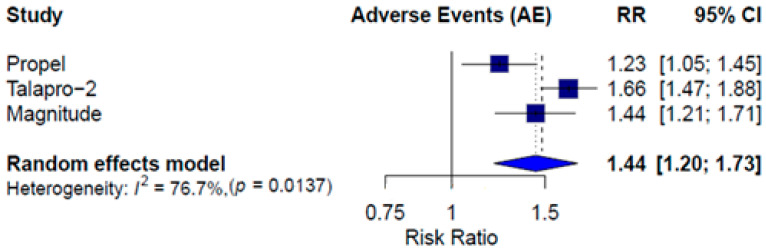
Forest plot estimating pooled RR of any AEs of PARP inhibitor combined therapy in the ITT population *. * In the MAGNITUDE trial, safety data were reported only for the HRRm population.

**Figure 7 pharmaceuticals-18-01015-f007:**
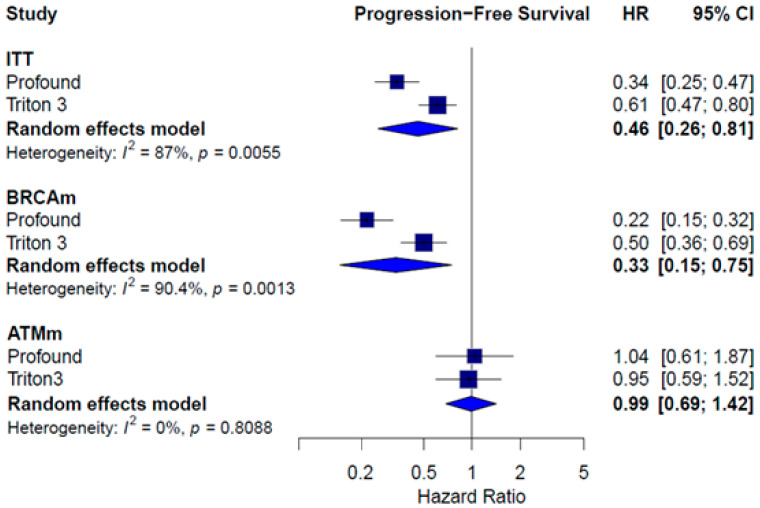
Forest plot estimating PFS of patients treated with iPARP monotherapy in comparison to AAP or enzalutamide or docetaxel.

**Figure 8 pharmaceuticals-18-01015-f008:**
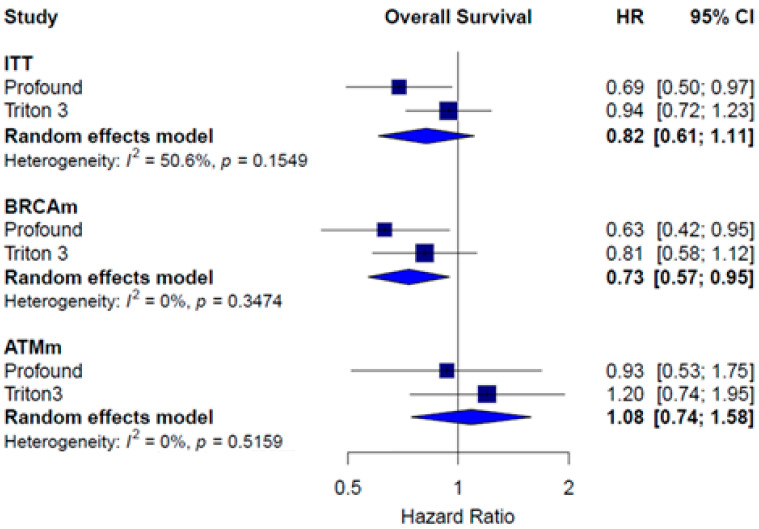
Forest plot estimating OS of patients treated with iPARP monotherapy compared to AAP, enzalutamide, or docetaxel.

**Figure 9 pharmaceuticals-18-01015-f009:**
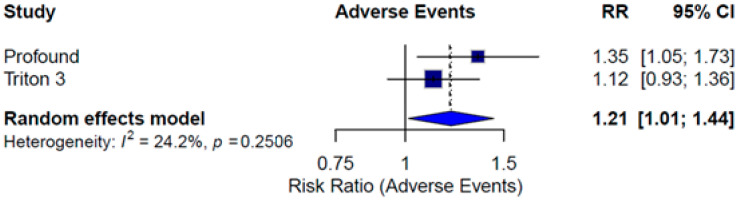
Forest plot estimating pooled RR of any AEs of iPARP monotherapy in the ITT population.

**Table 1 pharmaceuticals-18-01015-t001:** Characteristics of trials with iPARP in combination with ARSI in mCRPC patients.

Author (Year)/ Study	Clark N, 2022 PROpel [14]	Chi K, 2023 MAGNITUDE [13,25]	Agarwal N, 2023/2025; Fizazi K, 2025 TALAPRO-2 [15,16,17]
Population	Patients with mCRPC without prior systemic treatment for mCRPC	Patients with mCRPC without prior systemic treatment for mCRPC	Patients with mCRPC without prior systemic treatment for mCRPC
Previous therapies	Prior docetaxel treatment Olaparib: 97 (24.3%) Placebo: 98 (24.7%) Prior treatment with NHA Olaparib: 1 (0.3%) Placebo: 0	Androgen deprivation therapy Niraparib+AAP: 204 (96.2%) Placebo+AAP: 201 (95.3%) AR-targeted therapy for nmCRPC or mCSPC Niraparib+AAP: 8 (3.8%) Placebo+AAP: 5 (2.4%) Taxane chemotherapy for mCSPC Niraparib+AAP: 41 (19.3%) Placebo+AAP: 44 (20.9%) AAP (≤4 months) for mCRPC Niraparib+AAP: 50 (23.6%) Placebo+AAP: 48 (22.7%) Others # Niraparib+AAP: 52 (24.5%) Placebo+AAP: 58 (27.5%)	Previous docetaxel chemotherapy Talazoparib plus enzalutamide: 86 (21%) Placebo plus enzalutamide: 93 (23%) Previous treatment with novel hormonal therapy Talazoparib plus enzalutamide: 23 (6%) Placebo plus enzalutamide: 27 (7%) Abiraterone Talazoparib plus enzalutamide: 21 (5%) Placebo plus enzalutamide: 25 (6%) Orteronel Talazoparib plus enzalutamide: 2 (<1%) Placebo plus enzalutamide: 2 (<1%)
Intervention, *n*	Olaparib (300 mg BID) and AAP (1000 mg/day) *N* = 399 HRRm = 111 (27.8%) Non-HRRm = 279 (69.9%) BRCA1 = 9 (2.3%) BRCA2 = 38 (9.5%)	Niraparib (200 mg/day) and AAP (1000 mg/day) *N* = 335 HRRm = 212 (63.2%) Non-HRRm = 123 (36.7%) * * Futility analysis during the study’s development	Talazoparib (0,5 mg) and enzalutamide (160 mg/day) (*N* = 402) HRRm = 85 (21%) BRCA1/2 = 27 (7%) Non-HRRm = 317 (79%)
Comparator, *n*	AAP (1000 mg/day) and placebo (*N* = 397) HRRm = 115 (29%) Non-HRRm = 273 (68.8%) BRCA1 = 3 (0.8%); BRCA2 = 35 (8.8%)	AAP (1000 mg/day) and placebo (*N* = 335) HRRm = 211 (62.9%) Non-HRRm = 124 (37.1%) * * Futility analysis during the study’s development	Placebo plus enzalutamide (160 mg/day) (*N* = 404) HRRm = 84 (21%) BRCA1/2 = 32 (8%) Non-HRRm = 319 (79%)
Median PFS	ITT HR = 0.61; 95% CI 0.50 to 0.75 Subgroups *HRRm* HR = 0.50; 95% CI 0.34 to 0.73 *Non-HRRm* HR = 0.76; 95% CI 0.60 to 0.97 *BRCAm* HR = 0.23; 95% CI 0.12 to 0.44	Subgroups *HRRm* HR = 0.73; 95% CI 0.56 to 0.96 *Non-HRRm* HR = 1.09; 95% CI 0.75 to 1.58 *BRCAm* HR = 0.53; 95% CI 0.36 to 0.79	ITT HR = 0.67; 95% CI 0.55 to 0.81 Subgroups *HRRm* *Talapro-2 (cohort 1)* HR = 0.51; 95% CI 0.36 to 0.72 *Talapro-2 (cohort 2)* HR = 0.47; 95% CI 0.36 to 0.61 *Non-HRRm* *Talapro (cohort 1)* HR = 0.69; 95% CI 0.49 to 0.98 *BRCAm* *Talapro (cohort 1)* HR = 0.24; 95% CI 0.12 to 0.48
Median OS	ITT HR = 0.81; 95% CI 0.66 to 0.99 Subgroup *HRRm* HR = 0.66; 95% CI, 0.45 to 0.96 *Non-HRRm* HR = 0.89; 95% CI 0.70 to 1.14 *BRCAm* HR = 0.29; 95% CI 0.15 to 0.58	Subgroup *HRRm* HR = 0.94; 95% CI 0.65 to 1.36 *BRCAm* HR = 0.96; 95% CI 0.57 to 1.62	ITT HR = 0.80; 95% CI 0.66 to 0.96 Subgroups *HRRm* *Talapro-2 (cohort 1)* HR = 0.55; 95% 0.36 to 0.83 *Talapro-2 (cohort 2)* HR = 0.62; 95% CI 0.48 to 0.81 *Non-HRRm* *Talapro-2 (cohort 1)* HR = 0.78; 95% CI 0.58 to 1.05 *BCRAm* *Talapro-2 (cohort 1)* HR = 0.53; 95% CI 0.28 to 1.02 *Talapro-2 (cohort 2)* HR = 0.50; 95% CI 0.32 to 0.78

AAP: abiraterone acetate plus prednisone, BRCAm: BRCA mutation, CI: confidence interval, HR: hazard ratio, HRRm: Homologous Recombination Repair mutation, ITT: intention-to-treat, mCRPC: metastatic castration-resistant prostate cancer, mCSPC: metastatic castration-sensitive prostate cancer, NHA: Novel Hormonal Agent, nmCRPC: non-metastatic castration-resistant prostate cancer, OS: overall survival, PFS: progression-free survival. # Other therapies included steroids, immunotherapy (ipilimumab, sipuleucel-T), investigational drugs, and non-taxane chemotherapy (estramustine).

**Table 2 pharmaceuticals-18-01015-t002:** Characteristics of trials with iPARP monotherapy in mCRPC patients.

Author (Year)/ Study	De Bono, 2020 PROfound [10,11]	Fizazi, 2023 TRITON 3 [12]
Population	Patients with mCRPC with previous treatment with enzalutamide, abiraterone, or chemotherapy allowed Cohort A (*n* = 245) BRCA1, BRCA2, or ATM alteration Cohort B (*n* = 142) alteration in any of 12 prespecified genes (BRIP1, BARD1, CDK12, CHEK1, CHEK2, FANCL, PALB2, PPP2R2A, RAD51B, RAD51C, RAD51D, and RAD54L)	Patients with mCRPC with a BRCA1, BRCA2, or ATM alteration with previous treatment with enzalutamide, abiraterone, not chemotherapy
Previous therapies	Previous new hormonal agent *Cohort A and B* Enzalutamide only Olaparib: 105 (41%); control: 54 (41%) Abiraterone only Olaparib: 100 (39%); control: 54 (41%) Enzalutamide and abiraterone Olaparib: 51 (20%); control: 23 (18%) *Cohort A* Enzalutamide only Olaparib: 68 (42%); control: 40 (48%) Abiraterone only Olaparib: 62 (38%); control: 29 (35%) Enzalutamide and abiraterone Olaparib: 32 (20%); control: 14 (17%) Previous taxane use *Cohort A and B* Olaparib: 170 (66%); control: 84 (64%) Docetaxel only Olaparib: 115 (45%); control: 58 (44%) Cabazitaxel only Olaparib: 3 (1%); control: 0 Docetaxel and cabazitaxel Olaparib: 51 (20%); control: 26 (20%) Paclitaxel only Olaparib: 1 (<1%); control: 0 *Cohort A* Previous taxane use Olaparib: 106 (65%); control: 52 (63%) Docetaxel only Olaparib: 74 (46%); control: 32 (39%) Cabazitaxel only Olaparib: 2 (1%); control: 0 Docetaxel and cabazitaxel Olaparib: 29 (18%); control: 20 (24%) Paclitaxel only Olaparib: 1 (<1%); control: 0	Second-generation ARPI Abiraterone acetate Rucaparib: 150 (56%); control: 80 (59%) Apalutamide Rucaparib: 8 (3%); control: 1 (1%) Enzalutamide Rucaparib: 119 (44%); control: 61 (45%) Docetaxel for hormone-sensitive prostate cancer Rucaparib: 63 (23%); control: 28 (21%) Therapy for castration-resistant prostate cancer 0 Rucaparib: 48 (18%); control: 26 (19%) ≥1 Rucaparib: 222 (82%); control: 109 (81%) Assigned control medication Docetaxel Rucaparib: not applicable; control: 75 (56%) Abiraterone acetate Rucaparib: not applicable; control: 28 (21%) Enzalutamide Rucaparib: not applicable; control: 32 (24%)
Intervention, *n*	Olaparib (300 mg BID) Cohort A = 162 Cohort B = 94	Rucaparib (600 mg BID) *N* = 270
Comparator, *n*	Physician’s choice of enzalutamide (160 mg/day) or AAP (1000 mg/day) Cohort A = 8 Cohort B = 48	Physician’s choice of enzalutamide (160 mg/day) *N* = 32 (24%), or AAP (1000 mg/day) *N* = 28 (21%), or Docetaxel (75 mg/m^2^ every 3 weeks, up to a maximum of 10 cycles)
Median PFS and OS	Results for Cohort A and B: PFS (independent review) Olaparib: 5.8 months; control: 3.5 months HR = 0.49; 95% CI, 0.38–0.63; *p* < 0.001) OS Olaparib: 17.5 months; control: 14.3 months HR = 0.67; 95% CI 0.49–0.93; *p* = 0.006 Cohort A PFS (independent review) Olaparib: 7.4 months; control: 3.6 months HR = 0.34; 95% CI, 0.25 to 0.47; *p* < 0.001 OS Olaparib: 19.1 months; control: 14.7 months HR = 0.69; 95% CI 0.50–0.97; *p* = 0.006 Cohort B PFS (independent review) Olaparib: 4.8 months; control: 3.3 months HR 0.88	Results for ITT population: PFS (independent review) Rucaparib: 10.2 (8.3–11.2) months; control: 6.4 (5.6–8.2) months HR = 0.61; 95% CI, 0.47–0.80; *p* < 0.001 OS Rucaparib: 23.6 (19.7–25.0) months; control: 20.9 (17.5–24.4) months HR = 0.94; 95% CI, 0.72 to 1.23

AAP: abiraterone acetate plus prednisone, ARPI: Androgen Receptor Pathway Inhibitor, CI: confidence interval, HR: hazard ratio, ITT: intention-to-treat, mCRPC: metastatic castration-resistant prostate cancer, OS: overall survival, PFS: progression-free survival.

**Table 3 pharmaceuticals-18-01015-t003:** Grade ≥3 adverse events occurring in >2% of ITT patients with combined therapy with PARP inhibitors *.

Event	PROpel [14]		MAGNITUDE [13]		TALAPRO-2 [15]	
	Olaparib and AAP (*N* = 398)	Placebo and AAP (*N* = 396)	Niraparib and AAP (*N* = 212)	Placebo and AAP (*N* = 211)	Talazoparib and Enzalutamide (*N* = 398)	Placebo and Enzalutamide (*N* = 401)
Any	188 (47.2)	152 (38.4)	142 (67.0)	98 (46.4)	299 (75.1)	181 (45.1)
Anemia	60 (15.1) ^ǂ^	13 (3.3) ^ǂ^	63 (29.7)	16 (7.6)	185 (46.5)	17 (4.2)
Neutropenia			14 (6.6)	3 (1.4)	73 (18.3)	6 (1.5)
Fatigue	9 (2.3)	6 (1.5)	7 (3.3)	9 (4.3)	16 (4.0)	8 (2.0)
Thrombocytopenia			14 (6.6)	5 (2.4)	29 (7.3)	4 (1.0)
Back pain	3 (0.8)	4 (1.0)	51 (24.1)	2 (1.0)	10 (2.5)	4 (1.0)
Leukopenia			5 (2.4)	1 (0.5)	25 (6.3)	0
Decreased appetite	4 (1.0)	0	6 (2.8)	1 (0.5)	5 (1.3)	4 (1.0)
Fall			2 (0.9)	6 (2.8)	9 (2.3)	8 (2.0)
Asthenia			2 (0.9)	1 (0.5)	11 (2.8)	3 (0.7)
Hypertension	14 (3.5)	13 (3.3)	31 (14.6)	26 (12.3)	21 (5.3)	30 (7.5)
Lymphopenia					20 (5.0)	4 (1.0)
Hypokalemia			6 (2.8)	6 (2.8)		
Urinary tract infection	8 (2.0)	4 (1.0)				

AAP, abiraterone acetate with prednisone. Common Terminology Criteria for Adverse Events version 4.03 was used. ^ǂ^ Anemia included anemia, decreased hemoglobin level, decreased red blood cell count, decreased hematocrit level, erythropenia, macrocytic anemia, normochromic anemia, normochromic normocytic anemia, and normocytic anemia. * Values represent the number of reported grade ≥3 AEs in each study arm, and ‘0’ indicates that no events were reported. Blank cells correspond to data not reported in the original publication.

**Table 4 pharmaceuticals-18-01015-t004:** Grade ≥3 adverse events occurring in >2% of ITT patients with monotherapy with PARP inhibitors *.

Event	PROfound [10,11]		TRITON3 [12]	
	Olaparib (*N* = 256)	Control (*N* = 130)	Rucaparib (*N* = 270)	Control (*N* = 130)
Any, *n* (%)	130 (50.8)	49 (37.7)	161 (59.6)	69 (53.1)
Anemia, *n* (%)	55 (21.5)	7 (5.4)	64 (23.7)	1 (0.8)
Neuropathy ^‡^, *n* (%)			25 (9.3)	4 (3.1)
Neutropenia, *n* (%)	10 (3.9)	0	20 (7.4)	10 (7.7)
Thrombocytopenia, *n* (%)	9 (3.5)	0	16 (5.9)	0
Fatigue/asthenia, *n* (%)	7 (2.7)	7 (5.4)	19 (7.0)	12 (9.2)
Pneumonia, *n* (%)	6 (2.3)	1 (0.8)		
Dyspnea, *n* (%)	6 (2.3)	0	1 (0.4)	2 (1.5)
Vomiting, *n* (%)	6 (2.3)	1 (0.8)	2 (0.7)	1 (0.8)
Pulmonary embolism, *n* (%)	6 (2.3)	1 (0.8)		
Urinary tract infection, *n* (%)	4 (1.6)	5 (3.8)		
Hypertension, *n* (%)	3 (1.2)	3 (2.3)	16 (5.9)	7 (5.4)
Sepsis, *n* (%)	3 (1.2)	3 (2.3)		
Nausea, *n* (%)	3 (1.2)	0	7 (2.6)	1 (0.8)
Back pain, *n* (%)	2 (0.8)	2 (1.5)	9 (3.3)	5 (3.8)
Febrile neutropenia, *n* (%)			2 (0.7)	8 (6.2)

^‡^ Neuropathy includes neurotoxicity, paresthesia, peripheral motor neuropathy, peripheral neuropathy, peripheral sensory neuropathy, and polyneuropathy. * Values represent the number of reported grade ≥3 AEs in each study arm, and “0” indicates no events were reported. Blank cells correspond to data not reported in the original publication.

**Table 5 pharmaceuticals-18-01015-t005:** Summary of key outcomes (rPFS, OS, PFS2, and grade ≥3 AEs) across monotherapy and combination therapy by molecular subgroup.

Subgroup	Treatment	PFS HR [CI]	OS HR [CI]	PFS2 HR [CI]	Grade ≥3 AEs RR [CI]
Monotherapy	ITT	0.46 [0.26; 0.81]	0.82 [0.61; 1.11]		1.21 [1.01; 1.44]
	BRCAm	0.33 [0.15; 0.75]	0.73 [0.57; 0.95]		
	ATMm	0.99 [0.69; 1.42]	1.08 [0.74; 1.58]		
Combined therapy	ITT	0.64 [0.56; 0.74]	0.80 [0.70; 0.92]	0.77 [0.64; 0.91]	1.44 [1.20; 1.73]
	HRRm	0.55 [0.44; 0.68]	0.68 [0.55; 0.83]		
	Non-HRRm	0.81 [0.64; 1.03]	0.84 [0.70; 1.02]		
	BRCAm	0.33 [0.18; 0.58]	0.54 [0.34; 0.85]		

## Data Availability

The data are not available for public access but may be available from the corresponding author upon reasonable request.

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
