# Peer review of "PARP Inhibitors for Metastatic CRPC: More Answers than Questions, a Systematic Review and Meta-Analysis"

_pharmaceuticals, 2025, doi:10.3390/ph18071015_

Round 1
Reviewer 1 Report
Comments and Suggestions for Authors
Comments
The manuscript provides a comprehensive and timely synthesis of the efficacy and safety of PARPi-based therapies in mCRPC, with a strong focus on molecular subgroups. The systematic review and meta-analysis are methodologically sound, and the inclusion of recent trial data (e.g., TALAPRO-2 final OS results) adds significant value. Addressing the below comments will further enhance the clarity, rigor, and clinical applicability of the present manuscript.
- The objectives of the study are well-articulated in the abstract and introduction, focusing on evaluating the efficacy and safety of PARP inhibitors (PARPi) as monotherapy or in combination with ARSI in mCRPC. However, the specific research questions (e.g., which molecular subgroups benefit most, or how toxicity profiles influence clinical decision-making) could be more explicitly stated in the introduction to guide readers. Consider adding a concise list of research questions to strengthen the focus.
- The manuscript includes only phase III randomized controlled trials (RCTs), which is appropriate for high-quality evidence synthesis, but the rationale for excluding phase II trials (e.g., TOPARP-A, referenced in the introduction) is not discussed. Providing a brief justification for this exclusion would enhance transparency, especially since phase II trials like TOPARP-A provided foundational evidence for PARPi in mCRPC.
- In meta-analysis reports high heterogeneity (e.g., I² = 87.0% for PFS in PARPi monotherapy, I² = 70.5% for rPFS in BRCAm combination therapy). While the random-effects model is appropriately used, the discussion of potential sources of heterogeneity (e.g., differences in patient populations, prior treatments, or PARPi agents) is limited. Please expand the discussion to address clinical and methodological sources of heterogeneity and their implications for interpreting the results.
- Manuscript notes no significant benefit in non-HRRm patients for combination therapy (HR = 0.81; 95% CI: 0.64-1.03 for rPFS; HR = 0.84; 95% CI: 0.70-1.02 for OS). Although , the discussion does not explore potential reasons for this lack of benefit or whether specific non-HRRm subgroups might still respond (e.g., based on other molecular or clinical characteristics). Please include a deeper analysis of this subgroup to guide future research directions.
- In the safety analysis for combination therapy (Table 3) is comprehensive, but the manuscript notes that MAGNITUDE trial safety data were only reported for the HRRm population. This limitation could bias the pooled relative risk estimate for grade ≥3 adverse events (AEs). Clarify how this was addressed in the meta-analysis (e.g., sensitivity analysis) or discuss the potential impact of this limitation on the results.
- The quality assessment using the Cochrane Risk of Bias tool is well-described, but the manuscript states that monotherapy trials had an "unclear risk of performance bias" (Appendix B). More details on the specific aspects contributing to this unclear risk (e.g., lack of blinding or incomplete reporting) would strengthen the methodological rigor. Additionally, consider including a summary figure or table of the risk of bias assessment in the main text for accessibility.
- Discussion appropriately notes that clinical trial populations may not reflect real-world mCRPC patients (Section 3),but this point could be expanded to discuss specific factors (e.g., access to molecular testing, prior treatment patterns, or comorbidities) that might affect the generalizability of PARPi-based therapies. This would provide practical insights for clinicians.
- In the references section, a similar meta-analysis by Sayyid et al. (Ref [28]) and highlights the inclusion of updated TALAPRO-2 and MAGNITUDE data as a strength. However, a more detailed comparison with other published meta-analyses or systematic reviews on PARPi in mCRPC would help contextualize the findings. Please elaborate on how this study adds unique value to the existing literature.
- The conclusions emphasize the importance of molecular testing for optimizing PARPi therapy. While this is a critical point, the manuscript could provide more practical guidance on implementing molecular testing in clinical practice (e.g., recommended assays, timing of testing, or challenges in resource-limited settings). This would enhance the clinical relevance of the findings.
- Section 3 mentions FDA approvals for olaparib, rucaparib, talazoparib, and niraparib in specific mCRPC subgroups, but in the manuscript does not discuss whether these approvals align with the meta-analysis findings, particularly for non-HRRm patients where benefits were less clear. A brief discussion of the regulatory implications of the results would strengthen the manuscript.
- Statistical methods (Section 4) are well-described, but the rationale for choosing the DerSimonian and Laird random-effects model over other approaches (e.g., fixed-effects or Bayesian models) is not provided. Additionally, the use of Egger’s test for publication bias is mentioned, but results are not reported due to the small number of studies. Clarifying these methodological choices and their limitations would improve transparency.
- The forest plots (Figures 2-8) are clear and informative, but the manuscript could benefit from additional visual aids, such as a summary table comparing key outcomes (rPFS, OS, PFS2, and AEs) across monotherapy and combination therapy for each subgroup. This would facilitate quick reference for readers and enhance the manuscript’s accessibility.
- The manuscript is generally well-written but requires minor grammatical corrections and Some sentences are unclear and need rephrasing for better readability.
Based on above I would like to suggest the minor revision of present research work.
Comments on the Quality of English Language
The manuscript is generally well-written but requires minor grammatical corrections and Some sentences are unclear and need rephrasing for better readability.
Reviewer 2 Report
Comments and Suggestions for Authors
In this manuscript, the authors present a systematic review and meta-analysis evaluating the efficacy of PARP inhibitor-based therapies, either as monotherapy or in combination with androgen receptor signaling inhibitors, in patients with metastatic castration-resistant prostate cancer. While the overall review is well-supported, the following issues should be addressed:
- Figures 2–8: The contents in the first column overlap with the forest plots. Please adjust the formatting to improve readability and visual clarity.
- Tables 3 and 4: There are some blank and “0” values. Please clarify whether they represent the same in the table legends.
- Table 3: There are a few minor calculation errors or formatting inconsistencies. For example:
"142 (66.9)" should be corrected to "142 (67.0)";
"299 (75)" should be corrected to "299 (75.1)";
"3 (<1)" should be corrected to "3 (0.7)".
Please review and correct these values to ensure accuracy.
